# Quality Control of Targeted Plasma Lipids in a Large-Scale Cohort Study Using Liquid Chromatography–Tandem Mass Spectrometry

**DOI:** 10.3390/metabo13040558

**Published:** 2023-04-13

**Authors:** Akiyoshi Hirayama, Takamasa Ishikawa, Haruka Takahashi, Sanae Yamanaka, Satsuki Ikeda, Aya Hirata, Sei Harada, Masahiro Sugimoto, Tomoyoshi Soga, Masaru Tomita, Toru Takebayashi

**Affiliations:** 1Institute for Advanced Biosciences, Keio University, Tsuruoka 997-0052, Yamagata, Japan; 2Systems Biology Program, Graduate School of Media and Governance, Keio University, Fujisawa 252-0082, Kanagawa, Japan; 3Faculty of Environment and Information Studies, Keio University, Fujisawa 252-0082, Kanagawa, Japan; 4Department of Preventive Medicine and Public Health, Keio University School of Medicine, Shinjuku 160-8582, Tokyo, Japan; 5Institute of Medical Research, Tokyo Medical University, Shinjuku 160-0022, Tokyo, Japan

**Keywords:** targeted lipidomics, quality control, liquid chromatography–mass spectrometry, lipid, cohort study

## Abstract

High-throughput metabolomics has enabled the development of large-scale cohort studies. Long-term studies require multiple batch-based measurements, which require sophisticated quality control (QC) to eliminate unexpected bias to obtain biologically meaningful quantified metabolomic profiles. Liquid chromatography–mass spectrometry was used to analyze 10,833 samples in 279 batch measurements. The quantified profile included 147 lipids including acylcarnitine, fatty acids, glucosylceramide, lactosylceramide, lysophosphatidic acid, and progesterone. Each batch included 40 samples, and 5 QC samples were measured for 10 samples of each. The quantified data from the QC samples were used to normalize the quantified profiles of the sample data. The intra- and inter-batch median coefficients of variation (CV) among the 147 lipids were 44.3% and 20.8%, respectively. After normalization, the CV values decreased by 42.0% and 14.7%, respectively. The effect of this normalization on the subsequent analyses was also evaluated. The demonstrated analyses will contribute to obtaining unbiased, quantified data for large-scale metabolomics.

## 1. Introduction

Metabolomics is an *omics* science that analyzes hundreds of metabolites in biological samples. Recent improvements in this technology have enabled high-throughput and large-scale metabolomic studies. Metabolomics has been widely applied in epidemiological studies that include ≥1000 participants. These studies have led to the discovery of novel metabolic features and biomarkers for various chronic diseases, such as diabetes [1,2], cardiovascular disease [3,4], chronic kidney disease [5], Alzheimer’s disease [6], obesity [7], and blood pressure [8]. In addition to efficient data sharing and standardization, the Consortium of Metabolomics Studies (COMETS) was established to encourage large-scale collaboration of prospective cohort studies with human metabolome research [9].

Nuclear magnetic resonance (NMR) spectroscopy and mass spectrometry (MS) have been used in large-scale metabolomic studies. NMR is suitable for large-scale cohort studies involving long-term measurements because it is highly reproducible, and metabolites can be measured following simple pretreatment. However, NMR has relatively low sensitivity; therefore, only abundant metabolites can be profiled. In contrast, MS can measure a wide range of metabolites with high selectivity and sensitivity. It is impossible to analyze all metabolites using either method because of their wide variety of physical and chemical properties. Therefore, a combination of separation systems with MS, such as gas chromatography–MS (GC–MS), liquid chromatography–MS (LC–MS), and capillary electrophoresis–MS (CE–MS) are used according to the properties of the metabolites to be measured.

The Tsuruoka Metabolomics Cohort Study (TMCS), a cohort study of the Japanese population, was conducted in April 2012. This study used CE–MS to quantify charged metabolites in plasma and urine, and LC–MS was used to analyze lipids in plasma. The samples were collected from more than 10,000 registered participants, which required long-term analyses including multiple batches. The quality control (QC) of the CE–MS plasma data for 52 months resulted in coefficient of variation (CV) values of quantified metabolites of <30% for 85.1% of metabolites [10]. Applications of this dataset have been published for various diseases [11,12,13], physical activity [14], and food intake [15].

A strategy for obtaining high-quality LC–MS data for large-scale metabolomics has been proposed. Luo et al. developed a pseudo-targeted LC–MS method to improve the stability of large-scale metabolomic data [16]. This method included a blank wash step that eliminated the build-up of contaminants from the system and a postcalibration process using QC samples to correct signal drift among multiple batches. Consequently, the CV of 54% of the metabolite features was <15% in three independent batches. Brunius et al. proposed a new approach, including interbatch metabolite feature alignment and intrabatch cluster-based drift correction, to normalize multiple batch data from large-scale nontargeted LC–MS metabolomic data [17].

An approach for overcoming problems related to the conjunction of multiple batches of LC–MS-based lipidomic data was examined in this study. We selected 147 lipid species that are considered to be clinically important, such as bioactive lipids and lipid mediators. This approach could be used for various metabolome analyses, including large-scale cohort studies, although the original use was targeted at LC–MS lipidomics.

## 2. Materials and Methods

### 2.1. Study Population and Sample Collection

TMCS is a Japanese cohort study that started in April 2012 (Tsuruoka City, Yamagata Prefecture, Japan), involving 11,002 participants aged 35 to 74 years old [10,12,13,14,18]. Participants were recruited from among attendees of annual municipal or workplace health checkup programs held at four city sites at baseline (from April 2012 to March 2015). Written informed consent was obtained from all participants. This study was approved by the Medical Ethics Committee of the School of Medicine, Keio University (approval No. 20110264) and the corresponding regulatory agencies, and all experiments were performed in compliance with approved guidelines.

Blood samples were collected in the morning after 12 h of overnight fasting, and plasma samples were prepared using EDTA-2Na as an anticoagulant and immediately stored at 4 °C. The samples were centrifuged at 1500× *g* for 10 min at 4 °C within 3 h of sampling. The upper layer was stored at −80 °C until lipid extraction.

### 2.2. Extraction of Target Lipids

Plasma samples (100 μL) were mixed with 300 μL of methanol containing the following internal standards (IS): cholic acid-d5, 500 nmol/L; fatty acid (FA) 18:0-d3, 500 nmol/L; acylcarnitine 18:0-d3, 100 nmol/L; platelet-activating factor (PAF) 18:0-d4, 50 nmol/L; lysoPAF 18:0-d4, 50 nmol/L; lysophosphatidylcholine (LPC) 16:0-d3, 50 nmol/L; sphinganine d17:0, 50 nmol/L; sphingosine d17:1, 50 nmol/L; sphinganine 1-phosphate (1P) d17:0, 50 nmol/L; sphingosine 1P d17:1, 50 nmol/L; ceramide 1P d18:1-12:0, 50 nmol/L; glucosylceramide d18:1-12:0, 50 nmol/L; and lactosylceramide d18:1-12:0, 50 nmol/L. The mixtures were vortexed for 5 min and then centrifuged at 9100× *g* for 10 min at 20 °C. The supernatant (250 μL) was used for solid-phase extraction (SPE).

SPE cartridges (MonoSpin^®^ C18, GL Sciences, Tokyo, Japan) were conditioned with 300 μL of methanol (containing 0.1 vol% formic acid), followed by 300 μL of Milli-Q water (containing 0.1 vol% formic acid). Milli-Q water (350 μL containing 0.1 vol% formic acid), 100 μL of methanol (containing 0.1 vol% formic acid), and 250 μL of supernatant were mixed in SPE cartridges using pipetting and then centrifuged at 2000× *g* for 2 min at 20 °C. The cartridges were washed with 300 μL of Milli-Q water (containing 0.1 vol% formic acid), and the collection tube was changed. Lipids were eluted using 200 μL of methanol (containing 0.1 vol % formic acid). The extracts (100 μL) and methanol (10 μL) containing 1 μmol/L of an external standard (PS 17:0-14:1) were mixed in a glass vial and subjected to LC–MS/MS analysis.

### 2.3. Targeted Lipid Analysis

LC–MS/MS analysis was performed using an Agilent 1290 Infinity LC system (Agilent Technologies, Santa Clara, CA, USA) coupled to an AB Sciex QTRAP 5500 mass spectrometer with a turbo ion spray electrospray ionization (ESI) source (Sciex, Framingham, MA, USA). LC separation was performed using an Acquity UPLC HSS T3 column (2.1 × 50 mm, 1.8 μm; Waters, Milford, MA, USA). The mobile phase comprised 3:1:1 water:methanol:acetonitrile (*v*/*v*/*v*) with 5 mmol/L ammonium formate and 1 μmol/L EDTA (A) and isopropanol with 5 mmol/L ammonium formate and 1 μmol/L EDTA (B). The addition of a trace amount of EDTA is known to create chelates with residual metals in the LC–MS system, resulting in improved peak shapes for many metabolites [19]. The flow rate was 0.3 mL/min, and the following linear gradient was used: 0–5 min, 0–40% B; 5–7.5 min, 40–64% B; 7.5–12 min, 64% B; 12–12.5 min, 64–82.5% B; 12.5–15 min, 82.5–83.46% B; and 15–17.5 min, 83.46–97% B followed by equilibration with 0% B for 5 min. The injection volume was 8 μL, and the column temperature was maintained at 45 °C. Columns were replaced when the pressure exceeded 1.5-times the initial value.

ESI–MS/MS analysis was performed in positive/negative switching mode using the following source parameters: ion spray voltage, 4500/−4500 V; dry gas temperature, 300 °C; curtain gas, 30 psi; collision gas, 6 psi; ion source gas 1, 40 psi; and ion source gas 2, 80 psi.

The multiple reaction monitoring (MRM) settings were determined using flow injection analyses of commercially available compounds. The lipids belonging to a respective group were measured based on the conditions of the standard internal compounds, including product ions, collision energy, and cell exit potential. The MRM conditions of IS and corresponding lipids are summarized in Appendix A.

### 2.4. Method Validation

The developed analytical method, including linearity, accuracy, precision, recovery, and sensitivity, was validated according to the bioanalytical method validation guidance for industry (2018) issued by the U.S. Food and Drug Administration.

The IS mixture was diluted with methanol to prepare a 19-point calibration standard at 0.2–100,000 nmol/L to evaluate linearity. Each calibration standard was analyzed five times, and the mean area was used to prepare the calibration curve. Linearity was assessed by calculating the least-squares regression and was expressed using the coefficient of determination. The linear dynamic range was determined to be within ±20% accuracy of the calibration curve.

The concentration of limit of detection (C_LOD_) can be calculated using Equation (1):C_LOD_ = t_s_ × C_STD_ × S_STD_,(1)
where t_s_ is Student’s t-distribution factor for four degrees of freedom at the 99% confidence level (t_s_ = 3.747), C_STD_ is the minimum concentration of the standard in the dynamic range, and S_STD_ is the standard deviation of the peak area after five repeated injections of the standard at the minimum concentration.

Accuracy and precision were evaluated based on low, middle, and high calibration-curve concentrations. The following low, middle, and high concentrations of each standard were used: acylcarnitine 18:0-d3, 0.5, 20, 2000 nmol/L; ceramide 1P d18:1-12:0, 1, 50, 5000 nmol/L; cholic acid-d5, 5, 250, 10,000 nmol/L; FA 18:0-d3, 20, 500, 20,000 nmol/L; glucosylceramide d18:1-12:0, 2.5, 100, 5000 nmol/L; lactosylceramide d18:1-12:0, 10, 100, 1000 nmol/L; LPC 16:0-d3, 2.5, 100, 5000 nmol/L; PAF 18:0-d4, 0.5, 50, 5000 nmol/L; lysoPAF 18:0-d4, 2.5, 1000, 5000 nmol/L; sphinganine d17:0, 5, 100, 5000 nmol/L; sphinganine 1P d17:0, 2.5, 100, 5000 nmol/L; sphingosine d17:1, 2.5, 50, 2500 nmol/L; and sphingosine 1P d17:1, 25, 250, 5000 nmol/L. Accuracy was calculated as a percentage of the quantified value with respect to the theoretical value, and precision was determined as the relative standard deviation of five measurements. The extraction recovery test was performed using plasma samples spiked with the standards at the following concentrations: acylcarnitine 18:0-d3, 100 nmol/L; ceramide 1P d18:1-12:0, 50 nmol/L; cholic acid-d5, 500 nmol/L; FA 18:0-d3, 500 nmol/L; glucosylceramide d18:1-12:0, 50 nmol/L; lactosylceramide d18:1-12:0, 50 nmol/L; LPC 16:0-d3, 50 nmol/L; PAF 18:0-d4, 50 nmol/L; lysoPAF 18:0-d4, 50 nmol/L; sphinganine d17:0, 50 nmol/L; sphinganine 1P d17:0, 50 nmol/L; sphingosine d17:1, 50 nmol/L; and sphingosine 1P d17:1, 50 nmol/L.

### 2.5. Quantitative and Normalization Method of Metabolomic Profile

The concentration of each lipid was calculated using Equation (2) and the corresponding IS (Appendix A).
C_Lipid_ = A_Lipid_/A_IS_ × C_IS_(2)
where C_Lipid_ is the concentration of the lipid, C_IS_ is the concentration of the corresponding IS, A_Lipid_ is the peak area of the lipid, and A_IS_ is the peak area of the IS. The median value of each lipid was calculated from the quantitative values of the five QC samples measured in the same batch, and the relative value of each lipid in the sample was obtained by dividing by this value.

### 2.6. Statistical Analysis

Metabolomic profiles with and without normalization were analyzed using partial least squares-discriminant analysis (PLS-DA). The concentrations of lipids below the detection limit were substituted with half of the minimum value across all detected samples. Additionally, the relationship between the metabolites and age (each being ten years old) was analyzed. Both data were log_10_ scaled and transformed to Z-scores. PLS-DA using all data resulted in R^2^ values. The generalization ability (Q^2^) was assessed at the average value of 5 times 10-fold cross-validations with various random values. The PatternHunter function implemented in MetaboAnalyst with the Spearman correlation option was used to explore the monotonous increase or decrease in metabolites depending on age. Scaling and normalization, similar to those for PLS-DA, were used for this analysis. 

JMP Pro 14.1.0 (SAS Institute, Cary, NC, USA) and MetaboAnalyst (ver. 5.0) [20] were used for data analyses.

## 3. Results and Discussion

### 3.1. Method Validation

In large-scale metabolomic studies, quantified data are generally collected over long periods of time in multiple batches. During this period, one of the most important factors for obtaining stable and reliable data is the use of well-validated analytical methods. The method used in this study was validated using IS. Figure 1 shows the chromatograms of the 13 deuterated or odd-chain IS used in this study.

Table 1 shows the calibration curve results, R^2^ value of the coefficients of determination, lower limit of detection, and linear dynamic range for the IS. The calibration curves for all compounds were linear, with coefficients of determination of 0.979–0.994. The limits of detection determined from the standard deviation of the peak area at a minimum concentration in the dynamic range were between 0.41 and 29.17 nmol/L, sufficient to detect low concentrations of lipid components. It was also found that the linear dynamic range of the investigated compounds was between two and four orders of magnitude, which could correspond to a wide concentration range.

Accuracy, precision, and extraction recovery were also investigated (Table 2). The error between the quantitative and theoretical values at the three concentrations was within 20% for all examined compounds. The precision of the results for some compounds exceeded 30% at low concentrations. However, almost all of these were within 10% at medium and high concentrations. These results suggest that the analytical method developed in this study is sufficiently accurate and precise for lipid quantification. The accuracy and precision results at 19 standard calibration concentration points are summarized in Appendix A.

Before sample preparation, the extraction recoveries for lipids from human plasma were determined by adding a known amount of IS mixtures (the concentration of each standard is described in Section 2.4). Except for LPC, the recoveries for the tested compounds ranged from 100.2% to 111.5%, indicating that the extraction of lipids from plasma could be quantitatively performed. Although the extraction recovery of LPC was slightly worse (63.0%) than that of the others in this study, the effect of reduced recovery due to sample preparation can be compensated for by spiking with isotope-labeled standards.

### 3.2. Comparison of Analytical Results with and without Normalization

This study measured 10,833 samples, with each batch consisting of 40 samples. We prepared a pooled QC sample at the beginning of the study and aliquots of the same pooled QC sample were used across all batches for the entire study. QC samples were measured first and last, and for every ten samples; therefore, 5 QC samples were measured for each batch, and 1376 QC samples were analyzed in all batches. The mass spectrometer was autocalibrated every three months, with maintenance performed once a year. Finally, it took 1580 days to measure all samples. Figure 2 shows the normalization strategy adopted in this study. The median value of each lipid was calculated from the quantitative values of the five QC samples measured in the same batch, and the relative value of each lipid in the sample was obtained by dividing by this value. 

Figure 3 shows the relative areas of the QC samples. Each relative area was divided by the average of those in all QC samples, and, therefore, the horizontal bar at y = 1.0 was the ideal line without any variations. Figure 3A shows the variations in FA, including FA 18:0, FA 18:1, FA 18:2, and FA 22:6. Figure 3B shows the variations in lactosylceramide, including lactosylceramide d18:1-14:0, lactosylceramide d18:1-16:0, lactosylceramide d18:1-18:0, and lactosylceramide d18:1-24:1. FA 18:0 exhibited relatively small variations, whereas FA 22:6 exhibited relatively large variations. Figure 3C shows chromatograms of the FA. Compared to FA 18:0, the retention time (RT) of the FA 22:6 was significantly different from that of the IS and exhibited a larger variation. This trend was the same for lactosylceramide (Figure 3D). Lactosylceramide d18:1-14:0, whose RT was closest to that of the IS, exhibited quantified data close to 1.0. Additionally, lactosylceramide d18:1-18:0 and lactosylceramide d18:1-24:1, whose RT was significantly different from that of the IS, exhibited a large difference. This large difference indicates that the fluctuations of these peaks were different from those of the IS. Among the 147 metabolites in the QC samples, the CV among batches (inter-CV) and the CV in a batch (intra-CV) are shown in Figure 3E,F. The median values of the inter- and intra-CV were 17.8% and 10.6%, respectively, indicating that the variance among batches was smaller.

Figure 4 shows the effect of normalization using QC samples on the peaks in the sample data. Figure 4A,B show the quantified data for FA 18:1 before and after normalization. The data for FA 18:1 before normalization exhibited a horizontally flat trend, and most of the data were less than 50 μM. Although several data at the sample numbers around No. 4800 exhibited conspicuously high values over 100 μM (Figure 4A), all data were flat after normalization (Figure 4B). Figure 4C,D show the quantified data for lactosylceramide d18:1-16:0 before and after normalization, respectively. The quantified data exhibited larger fluctuations than that of FA 18:1. These fluctuations did not follow a random trend, but the curves showed some patterns, for example, a gradual increase between samples No. 1 and 4000 (Figure 4C). In addition, abrupt changes were observed at around 4000, 7000, and 10,000 chromatographic runs, which correspond to annual instrument maintenance. After normalization, these patterns disappeared, and all data showed a horizontally flat trend (Figure 4D). The median value of the inter-CV before normalization was 20.8% (Figure 4E). After normalization, these values decreased to 14.7% (Figure 4F). The median value of the intra-CV before normalization was 44.3% and decreased to 42.0% after normalization.

### 3.3. Data Analysis

The effect of normalization on the subsequent statistical analyses was analyzed. Figure 5 shows the relationship between lipid profile and gender, and Figure 5A,B show the score plots of PLS-DA using the data without and with normalization, respectively. The analyses using the data without and with normalization using up to five components resulted in R^2^ = 0.859, Q^2^ = 0.856 ± 4.49×10^−5^ and R^2^ = 0.859, Q^2^ = 0.856 ± 1.30 × 10^−4^, respectively. The metabolites showing high variable importance in projection (VIP) scores within the top 20 without and with normalized data are shown in Figure 5C and Figure 5D, respectively. Both results include testosterone, one of the androgens, which consistently showed the largest VIP value. Acylcarnitine 20:4, FA 12:0, FA 14:1, FA 18:3, FA 12:1, lactosylceramide d18:1-14:0, acylcarnitine 20:3, and acylcarnitine 20:5 were included within the top ten VIP scores, although the orders were slightly different among them. As an inconsistent result, progesterone and LPC 20:2 were included in the analysis without normalized data, and glucosylceramide d18:1-18:0 was included in the analysis with normalized data.

The effect of the lipidomic profile on age was analyzed using PLS-DA, and the score plots are shown in Figure 6A,B. PLS-DA resulted in R^2^ = 0.521, Q^2^ = 0.515 ± 3.73 × 10^4^ using the data without normalization, and R^2^ = 0.527, Q^2^ = 0.519 ± 1.89 × 10^−4^ using the data with normalization. The VIP scores are shown in Figure 6C,D. Progesterone levels declined with age, and all other metabolites were included. FA 20:5 and acylcarnitine 20:5 were ranked first and second, respectively. FA 22:6 and acylcarnitine 22:5 were also consistently included, although the order differed. Lactosylceramide 18:0 was included in the data without normalization, whereas LPE 22:6, LPC 20:5, and LPC 24:0 were included in the data with normalization.

The PatternHunter function was used to identify metabolites that showed monotonous increases and decreases with age. The analyzed results using the data with and without normalization are shown in Figure 6E,F, respectively. FA 20:5 consistently showed the highest positive absolute correlation values in both results. Acylcarnitine 22:5, FA 22:6, and acylcarnitine 20:5 were included in the data without normalization. The negative correlation included progesterone and lysoPAF 18:0. Four metabolites of lactosylceramide and four metabolites of glucosylceramide were included. Only glucosylceramide d18:1-22:2 was included in the normalized data.

## 4. Conclusions

A method for the long-term stable measurement of lipid components contained in the plasma of more than 10,000 samples was developed in this study. Long-term quantification of stability using IS representing each lipid group alone is inadequate. In particular, lipids with a larger distance from the IS in the time dimension show large fluctuations. To minimize this problem, a method for correcting the median value of QC samples measured multiple times in the same batch was examined. Using this method, the interbatch CV decreased from 20.8% (before correction) to 14.7% (after correction). Several lipids were also found to be correlated with age and gender. 

In this study, all samples were analyzed using a single instrument. The usefulness of this method should be verified in the future by examining multiple apparatus, apparatus from various vendors, and multiple laboratories.

In conclusion, the correction method used in this study is versatile for various metabolome analyses, even in the long-term measurement of multiple samples, such as cohort studies.

## Figures and Tables

**Figure 1 metabolites-13-00558-f001:**
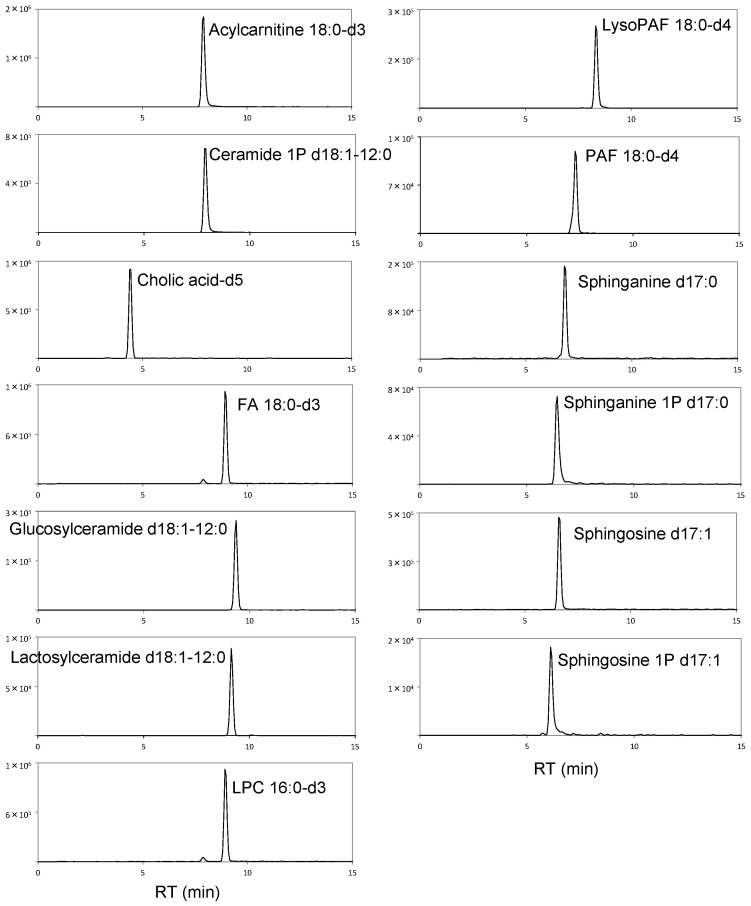
Chromatograms of the 13 compounds used for IS. X- and Y-axes indicate the retention time (RT; min) and intensity, respectively.

**Figure 2 metabolites-13-00558-f002:**
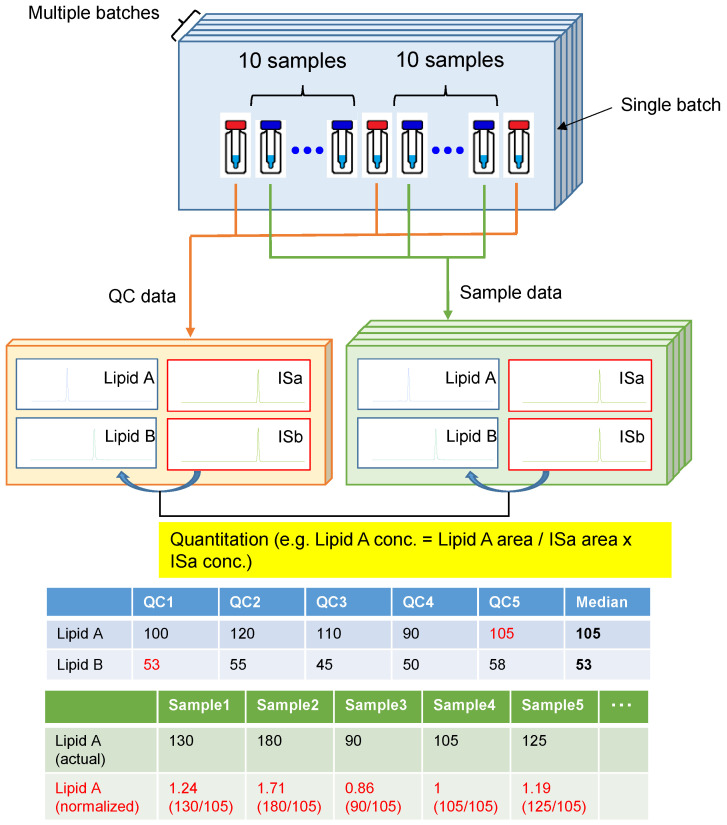
The quantitative and normalization method adopted in this study.

**Figure 3 metabolites-13-00558-f003:**
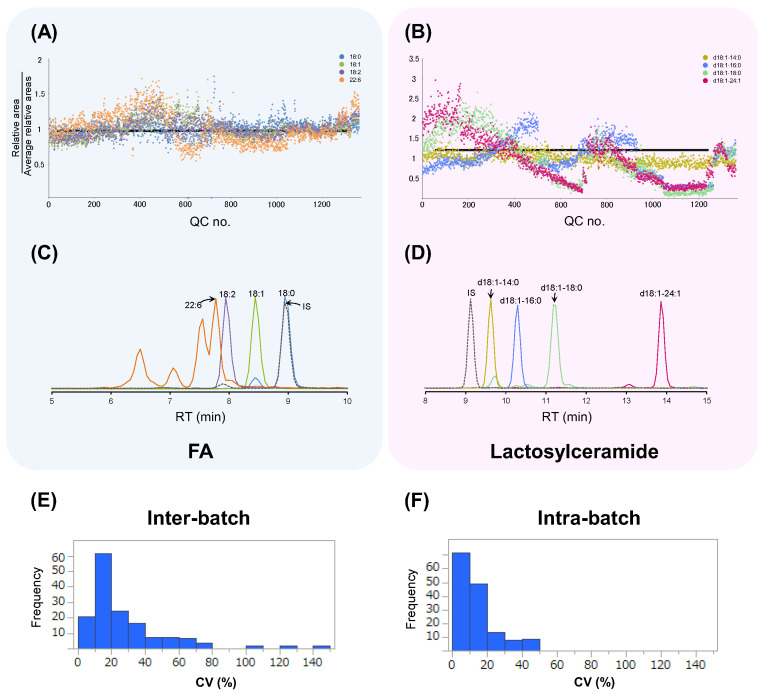
Fluctuation of the relative peak areas in the QC samples from multiple batches. (**A**,**C**) The FA data. (**B**,**D**) The lactosylceramide data. (**A**,**B**) Fluctuation of the relative peak area; that is, the peak area divided by the corresponding IS peak area. The X-axis indicates the QC sample number. (**C**,**D**) Representative chromatograms. (**E**) Histogram of the interbatch data. (**F**) Histogram of the intrabatch data. (**E**,**F**) The X- and Y-axes indicate the CV values and frequencies, respectively, of 147 metabolites.

**Figure 4 metabolites-13-00558-f004:**
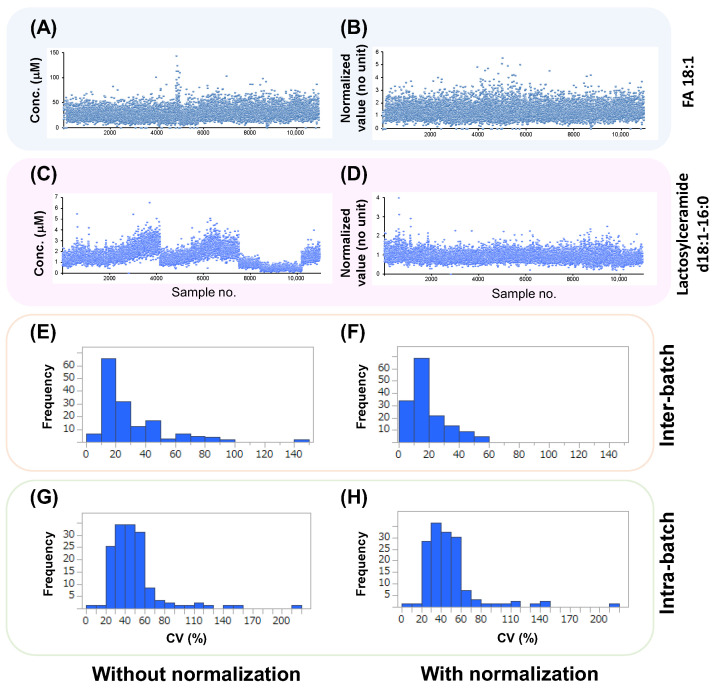
Comparison of the quantified values of the two metabolites without and with normalization. (**A**,**B**) The FA 18:1 data. (**C**,**D**) The lactosylceramide d18:1-16:0 data. (**E**,**F**) Histograms of the interbatch data. (**G**,**H**) Histograms of the intrabatch data. (**E**–**H**) The X- and Y-axes indicate the CV values and frequencies, respectively, of 147 metabolites.

**Figure 5 metabolites-13-00558-f005:**
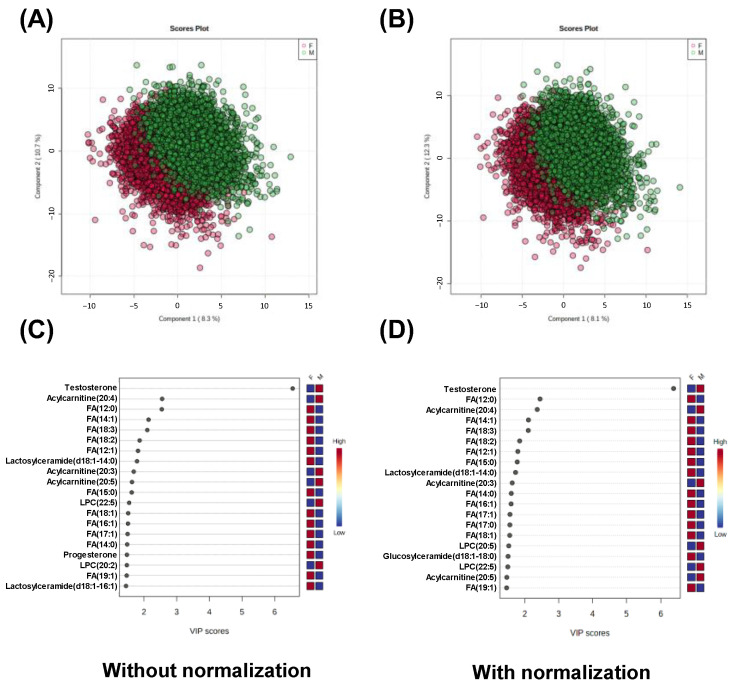
The relationship between gender and metabolomic profile using PLS-DA. (**A**,**B**) Score plots. (**C**,**D**) VIP scores. (**A**,**C**) Analytical results using the data without normalization and (**B**,**D**) analytical results using the data with normalization.

**Figure 6 metabolites-13-00558-f006:**
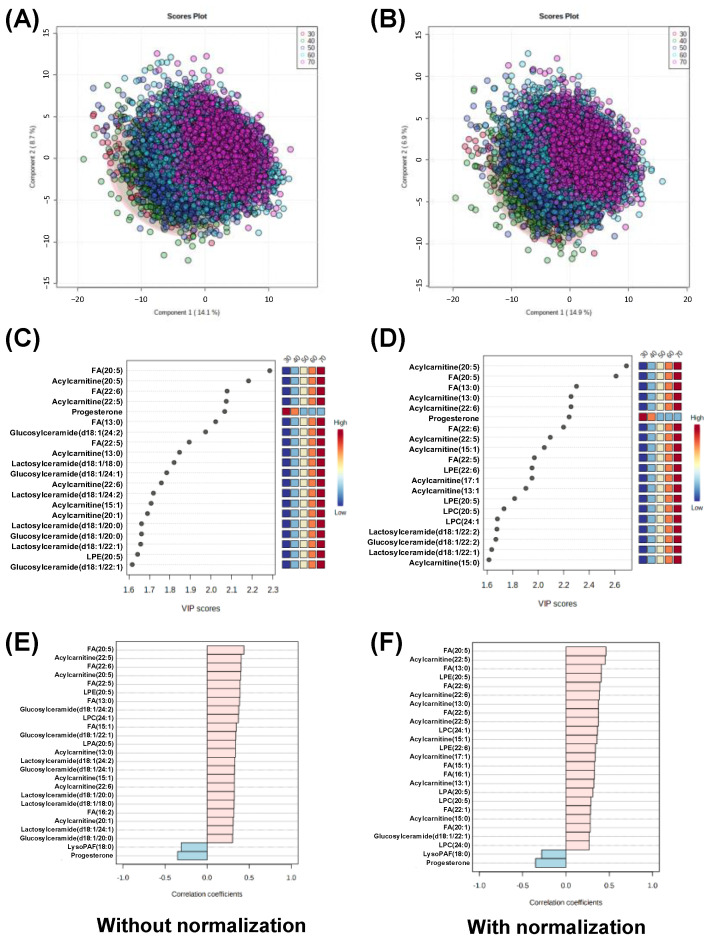
The relationship between ages and metabolomic profile using PLS-DA and PatternHunter. (**A**,**B**) Score plots. (**C**,**D**) VIP scores. (**E**,**F**) Correlations produced using PatternHunter. (**A**,**C**,**E**) Analytical results using the data without normalization and (**B**,**D**,**E**) analytical results using the data with normalization.

**Table 1 metabolites-13-00558-t001:** Linearity, sensitivity, and dynamic range of internal standard (IS) in the liquid chromatography–mass spectrometry (LC–MS/MS) method.

Compound	Calibration Curve	R^2^ Value	Limit of Detection (nmol/L)	Linear Dynamic Range (μmol/L)
Acylcarnitine 18:0-d3	y = (3.81 × 10^7^)x − 1.81 × 10^3^	0.994	0.41	0.0005–2
Ceramide 1P d18:1-12:0	y = (6.51 × 10^6^)x − 1.68 × 10^2^	0.988	1.13	0.001–5
Cholic acid-d5	y = (3.54 × 10^6^)x + 1.86 × 10^3^	0.994	3.48	0.005–10
FA 18:0-d3	y = (1.94 × 10^6^)x + 3.36 × 10^4^	0.979	23.05	0.02–20
Glucosylceramide d18:1-12:0	y = (1.09 × 10^7^)x − 3.16 × 10^3^	0.991	2.28	0.0025–5
Lactosylceramide d18:1-12:0	y = (3.87 × 10^6^)x − 4.48 × 10^3^	0.991	5.09	0.01–1
LPC 16:0-d3	y = (7.51 × 10^6^)x − 1.40 × 10^3^	0.991	2.03	0.0025–5
PAF 18:0-d4	y = (2.18 × 10^7^)x − 7.87 × 10^2^	0.990	0.45	0.0005–5
LysoPAF 18:0-d4	y = (1.21 × 10^7^)x − 3.45 × 10^3^	0.991	1.87	0.0025–5
Sphinganine d17:0	y = (6.41 × 10^6^)x + 7.79 × 10^3^	0.985	4.90	0.005–5
Sphinganine 1P d17:0	y = (3.93 × 10^6^)x − 1.30 × 10^3^	0.983	2.21	0.0025–5
Sphingosine d17:1	y = (2.12 × 10^7^)x + 6.83 × 10^3^	0.991	1.12	0.0025–2.5
Sphingosine 1P d17:1	y = (9.83 × 10^5^)x − 5.92 × 10^3^	0.983	29.17	0.025–5

**Table 2 metabolites-13-00558-t002:** Accuracy and precision of the IS in the LC–MS/MS method.

Compound	Low (n = 5, %)		Middle (n = 5, %)		High (n = 5, %)		Extraction Recovery (n = 3, %)
	Accuracy	Precision	Accuracy	Precision	Accuracy	Precision	
Acylcarnitine 18:0-d3	101.2	22.1	95.6	9.3	99.5	1.4	100.2 ± 10.0
Ceramide 1P d18:1-12:0	96.6	30.3	92.8	3.9	102.8	3.3	103.2 ± 8.1
Cholic acid-d5	98.3	18.6	105.0	4.4	102.2	5.6	111.1 ± 10.3
FA 18:0-d3	101.0	30.8	96.3	4.8	93.8	2.3	104.0 ± 7.4
Glucosylceramide d18:1-12:0	104.3	24.4	90.3	5.4	109.6	4.5	104.5 ± 12.1
Lactosylceramide d18:1-12:0	106.1	13.6	92.8	5.2	110.5	5.5	110.4 ± 8.0
LPC 16:0-d3	99.8	21.7	106.3	3.9	86.7	1.7	63.0 ± 3.8
PAF 18:0-d4	97.6	23.9	104.5	5.6	80.0	2.5	105.2 ± 8.8
LysoPAF 18:0-d4	97.4	19.9	100.1	10.7	85.5	3.9	106.9 ± 3.0
Sphinganine d17:0	99.3	26.2	103.3	7.0	87.1	4.8	102.9 ± 10.2
Sphinganine 1P d17:0	112.9	23.6	94.0	2.9	118.0	3.1	103.8 ± 5.4
Sphingosine d17:1	93.9	12.0	105.2	9.1	86.3	2.1	111.5 ± 5.9
Sphingosine 1P d17:1	109.5	31.1	92.7	5.1	112.1	2.3	102.6 ± 12.8

## Data Availability

The data presented in this study are available upon request from the corresponding author. The data are not publicly available to prevent misuse.

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
