# Peer review of "Quality Control of Targeted Plasma Lipids in a Large-Scale Cohort Study Using Liquid Chromatography–Tandem Mass Spectrometry"

_metabolites, 2023, doi:10.3390/metabo13040558_

Round 1

Reviewer 1 Report

This manuscript submitted by Hirayama et al. presents a targeted lipid profiling study on plasma samples collected from over 11,000 participants in the Tsuruoka Metabolomics Cohort Study (TMCS). The study design, validation and normalization techniques used over several batches in this long-term study based on a targeted lipid method using LC-MS/MS. The targeted lipid panel included acylcarnitines, modified sphingolipids, bile acids, fatty acids and lysophospholipids. Based on these lipids, the authors used PLS-DA as a model to determine correlations with sex and age of the cohort.  

-       While the authors presented an overall well study design and a unique approach for normalization in longitudinal studies, a few questions need to be addressed:

-       Why were these specific lipids targeted for this study? There are other classes of lipids that are very abundant and usually profiled in many lipidomic studies such as: phospholipids (phosphatidylcholines, phosphatidylethanolamines etc.), ceramides, sphingomyelin, diacylglycerols, triacylglycerols etc. Why were these excluded and not in the panel? The authors should briefly explain this in the main paper.

-       Line 100: Please provide details about the SPE cartridges used (Vendor, Part#)

-       Why did the authors choose to use SPE and not a simple Liquid-Liquid Extraction? Presumably the SPE helps enhance the signal of low abundance lipids (some of which are in the panel) while removing larger abundance lipids (i.e. phopsholipids, cholesterol). Again, this goes back to why these specific lipids were chosen as part of the panel (see first point).

-       For evaluating accuracy and precision, were the low, medium and high concentrations spiked into specific plasma samples or in QCs? What were the concentrations at the low, medium and high levels? How many replicates were performed per concentration? If more than one replicate was performed, the authors should provide a standard deviation along with the reported values in Table 2. This should also be done for the extraction recovery values reported.

-       Were aliquots of the same pooled QC sample used across all batches for the entire study? Or was a longitudinal QC sample used? The authors should be more clear about this in the manuscript.

-       Can the authors elaborate on the specifics of how the longitudinal studies were performed? How often were the instruments calibrated? How many days did it take to perform the entire study? Were the injection sequences randomized?

-       For statistical analysis, was imputation performed? If so, how were missing values imputed?

-       The normalization strategy presented by the authors using the median value of the QC samples in the same batch seems like a reasonable approach to use for long-term studies. Has this approach been compared to other normalization techniques/batch-correction methods used in other studies? It would be nice to compare how this method compares to other approaches, considering this normalization strategy is mathematically more simple and does not require extensive algorithms and complex coding like other batch-correction methods.

-       While the results and discussion elaborated on the technical aspects of this longitudinal study such as method validation, explanations of some of the candidate lipids that were correlated with sex and age, such as testosterone and progesterone respectively, were lacking. Are these findings consistent with other studies that evaluated age and sex based on these lipids? Were these lipids expected to be correlated with sex and age? If so, then why?

Author Response

This manuscript submitted by Hirayama et al. presents a targeted lipid profiling study on plasma samples collected from over 11,000 participants in the Tsuruoka Metabolomics Cohort Study (TMCS). The study design, validation and normalization techniques used over several batches in this long-term study based on a targeted lipid method using LC-MS/MS. The targeted lipid panel included acylcarnitines, modified sphingolipids, bile acids, fatty acids and lysophospholipids. Based on these lipids, the authors used PLS-DA as a model to determine correlations with sex and age of the cohort.  

-       While the authors presented an overall well study design and a unique approach for normalization in longitudinal studies, a few questions need to be addressed:

-       Why were these specific lipids targeted for this study? There are other classes of lipids that are very abundant and usually profiled in many lipidomic studies such as: phospholipids (phosphatidylcholines, phosphatidylethanolamines etc.), ceramides, sphingomyelin, diacylglycerols, triacylglycerols etc. Why were these excluded and not in the panel? The authors should briefly explain this in the main paper.

-       Why did the authors choose to use SPE and not a simple Liquid-Liquid Extraction? Presumably the SPE helps enhance the signal of low abundance lipids (some of which are in the panel) while removing larger abundance lipids (i.e. phopsholipids, cholesterol). Again, this goes back to why these specific lipids were chosen as part of the panel (see first point).

In this study, we focused on lipid species that are considered to be clinically important, such as bioactive lipids and lipid mediators. Since these lipids are generally in low abundance and cannot be sufficiently recovered by ordinary liquid-liquid extraction, we adopted a method using SPE. We have added the following sentences in the main text.

P2 Line 73

We selected 147 lipid species that are considered to be clinically important, such as bioactive lipids and lipid mediators.

-       Line 100: Please provide details about the SPE cartridges used (Vendor, Part#)

According to the reviewer’s suggestion we have added the details about the SPE cartridges used in this study.

-       For evaluating accuracy and precision, were the low, medium and high concentrations spiked into specific plasma samples or in QCs? What were the concentrations at the low, medium and high levels? How many replicates were performed per concentration? If more than one replicate was performed, the authors should provide a standard deviation along with the reported values in Table 2. This should also be done for the extraction recovery values reported.

Accuracy and precision were evaluated using calibration standard. The concentrations at low, middle, and high levels were described in the main text (P4, Line 150). We have added the values of relative standard deviation of individual standard in Table 2.

-       Were aliquots of the same pooled QC sample used across all batches for the entire study? Or was a longitudinal QC sample used? The authors should be more clear about this in the manuscript.

According to the reviewer’s suggestion we have added the following sentence in the main text.

P7 Line 226

We prepared a pooled QC sample at the beginning of the study and an aliquots of the same pooled QC sample were used across all batches for the entire study.

-       Can the authors elaborate on the specifics of how the longitudinal studies were performed? How often were the instruments calibrated? How many days did it take to perform the entire study? Were the injection sequences randomized?

We performed auto-calibration of MS every three month, and maintenance was performed once a year. Totally 1,580 days were needed to complete this study. In this study, sample was randomized, then injection sequences were not randomized. We have added the following sentence in the main text.

P7 Line 224

The mass spectrometer was auto-calibrated every three months, with maintenance performed once a year. Finally, it took 1,580 days to measure all samples.

-       For statistical analysis, was imputation performed? If so, how were missing values imputed?

In this study, for lipids which concentration was below the detection limit were substituted with half of the minimum value across all detected samples. We have added the following sentence in the main text.

P4 Line 176

The concentrations of lipids below the detection limit were substituted with half of the minimum value across all detected samples.

-       The normalization strategy presented by the authors using the median value of the QC samples in the same batch seems like a reasonable approach to use for long-term studies. Has this approach been compared to other normalization techniques/batch-correction methods used in other studies? It would be nice to compare how this method compares to other approaches, considering this normalization strategy is mathematically more simple and does not require extensive algorithms and complex coding like other batch-correction methods.

Thank you for the comment. We also wanted to perform a comparison with the standardization methods used in other studies, but many of the normalizations reported so far rely on QCs being measured frequently and also differing between QCs. Although it was practical for measuring hundreds of samples, it was not applicable to cohort studies measuring tens of thousands of samples, so we did not perform comparisons this time.

-       While the results and discussion elaborated on the technical aspects of this longitudinal study such as method validation, explanations of some of the candidate lipids that were correlated with sex and age, such as testosterone and progesterone respectively, were lacking. Are these findings consistent with other studies that evaluated age and sex based on these lipids? Were these lipids expected to be correlated with sex and age? If so, then why?

Testosterone is known as a male hormone, and progesterone is known as a female hormone, both of which are known to be high in men and women. We also know that this amount decreases with age. In our results, these trends did not change with or without normalization, suggesting that our normalization method is an appropriate method without eliminating the effects of characteristic lipids.

Reviewer 2 Report

First of all, I would like to congratulate the authors for their valuable work. The topic discussed is of definite interest in the field of metabolite analysis by LC/MS of large sample cohorts. The authors come to the important conclusion that quantification of long-term stability using IS alone is inadequate. Instead, it seems that a correction method based on the median value of QC samples measured several times in the same batch is more effective.

However, I have some observations/suggestions:
1) Title, abstract and introduction
The title carries the term "Plasma Lipid Profiles," and the abstract refers to the concept of "metabolomics." In addition, "metabolomics" and "Lipidomics" are also among the keywords. All these terms refer to an untargeted analytical approach, when in fact the manuscript reports the outcome of a clearly targeted study. I therefore suggest that the title, abstract and keywords be changed so that it is more relevant to the subject matter.

2) Materials and Methods.

I would replace the titles of the paragraphs "Lipid extraction" and Lipid Analysis" with more relevant ones such as "extraction of target metabolites." In fact, the method used excludes a large portion of plasma lipids such as glycerophospholipids and glycerolipids that normally constitute (along with cholesterol and cholesterol esters) the majority of plasma lipids.

These same paragraphs are missing some information that I would recommend adding so that any reader can repeat the experiment: for example, I would indicate the origin of the extraction method used: is the method an adaptation of one previously published? If yes, the reference should be given; if no, the rationale for adopting the methodology used should be indicated. The type and brand of SPE cartridges is missing. Finally, I find the addition of EDTA in the eluents very unusual. Perhaps the function performed by this additive should be explained. Does it enhance the ionization of certain metabolites? if so which ones?

Has the same column been used consistently for more than 10 thousand analyses? If yes, has its performance been measured from time to time? I also imagine that the mass spectrometer was calibrated before each batch. Is this the case? These seem trivial and obvious concepts but since this is a comparison of batches I think it is important to provide the reader with this information.

Method validation: I recommend that the supplemental include a table with data from the 19 concentration levels for each standard, the mean and standard deviation of the replicates of the calibration lines. A chromatogram with the peaks of the standards all together to give an idea of the effectiveness of the gradient would also be useful.

3) Results and discussion
Method validation: since these are calibration straight lines that span multiple orders of magnitude of concentration, the homo- or hetero-schedasticity of the calibration straight lines should be checked in addition to R2. R2 in these cases is not sufficient. The range of linearity could narrow further if only the range where the calibration line is homoschedastic is taken.
Figure 4C: What events correspond to the abrupt changes after 4000, 7000 and 1000 chromatographic runs? Change of column? Change of operator?  Cleaning of source or capillary? I would try to explain in the text. Moreover, how do you explain the absence of the same fluctuations in Figure 4A?

4) Conclusion.
I would recommend including a paragraph mentioning the limitations of this study, such as the fact that, from what I understand, only one analytical apparatus was used. In other words, it will be interesting in the future to involve more than one laboratory and/or more than one instrumentation. Indeed, in a clinical application perspective, it will be necessary to correct the values found between different laboratories.

Author Response

First of all, I would like to congratulate the authors for their valuable work. The topic discussed is of definite interest in the field of metabolite analysis by LC/MS of large sample cohorts. The authors come to the important conclusion that quantification of long-term stability using IS alone is inadequate. Instead, it seems that a correction method based on the median value of QC samples measured several times in the same batch is more effective.

However, I have some observations/suggestions:
1) Title, abstract and introduction
The title carries the term "Plasma Lipid Profiles," and the abstract refers to the concept of "metabolomics." In addition, "metabolomics" and "Lipidomics" are also among the keywords. All these terms refer to an untargeted analytical approach, when in fact the manuscript reports the outcome of a clearly targeted study. I therefore suggest that the title, abstract and keywords be changed so that it is more relevant to the subject matter.

Thank you for the suggestion. We have revised the title and keywords as follows.

Title: Quality Control of Targeted Plasma Lipids in a Large-scale Cohort Study Using Liquid Chromatography-Tandem Mass Spectrometry

Keywords: Targeted Lipidomics; Quality control; Liquid chromatography-mass spectrometry; Lipid; Cohort-study

2) Materials and Methods.

I would replace the titles of the paragraphs "Lipid extraction" and Lipid Analysis" with more relevant ones such as "extraction of target metabolites." In fact, the method used excludes a large portion of plasma lipids such as glycerophospholipids and glycerolipids that normally constitute (along with cholesterol and cholesterol esters) the majority of plasma lipids.

Thank you for the comment. According to the reviewer’s suggestion we have revised the title as follows. “Lipid extraction” to “Extraction of target lipids”. “Lipid analysis” to “Targeted Lipid analysis”.

These same paragraphs are missing some information that I would recommend adding so that any reader can repeat the experiment: for example, I would indicate the origin of the extraction method used: is the method an adaptation of one previously published? If yes, the reference should be given; if no, the rationale for adopting the methodology used should be indicated. The type and brand of SPE cartridges is missing. Finally, I find the addition of EDTA in the eluents very unusual. Perhaps the function performed by this additive should be explained. Does it enhance the ionization of certain metabolites? if so which ones?

According to reviewer’s suggestion we have added the following sentences in the main text.

P3 Line 101

SPE cartridges (MonoSpin® C18, GL Sciences, Tokyo, Japan) were conditioned with 300 μL of methanol (containing 0.1 vol % formic acid), followed by 300 μL of Milli-Q water (containing 0.1 vol % formic acid).

P3 Line 119

The addition of a trace amount of EDTA is known to create chelates with residual metals in the LC-MS system, resulting in improved peak shapes for many metabolites [ref. Myint et al., Anal. Chem. 2009].

Has the same column been used consistently for more than 10 thousand analyses? If yes, has its performance been measured from time to time? I also imagine that the mass spectrometer was calibrated before each batch. Is this the case? These seem trivial and obvious concepts but since this is a comparison of batches I think it is important to provide the reader with this information.

We performed auto-calibration of MS every three month, and maintenance was performed once a year. Columns were replaced when they exceeded 1.5 times the initial pressure. We have added following sentence in the main text.

P3 Line 124

Columns were replaced when the pressure exceeded 1.5-times the initial value.

P7 Line 230

The mass spectrometer was auto-calibrated every three months, with maintenance performed once a year. Finally it took 1,580 days to measure all samples.

Method validation: I recommend that the supplemental include a table with data from the 19 concentration levels for each standard, the mean and standard deviation of the replicates of the calibration lines. A chromatogram with the peaks of the standards all together to give an idea of the effectiveness of the gradient would also be useful.

Thank you for the comment. We have added the information of accuracy and precision at 19 points of standard calibration concentration in supplemental table2.

3) Results and discussion
Method validation: since these are calibration straight lines that span multiple orders of magnitude of concentration, the homo- or hetero-schedasticity of the calibration straight lines should be checked in addition to R2. R2 in these cases is not sufficient. The range of linearity could narrow further if only the range where the calibration line is homoschedastic is taken.
Figure 4C: What events correspond to the abrupt changes after 4000, 7000 and 1000 chromatographic runs? Change of column? Change of operator?  Cleaning of source or capillary? I would try to explain in the text. Moreover, how do you explain the absence of the same fluctuations in Figure 4A?

Thank you for valuable comments. In the calibration curve created this time, each point is weighted. Although we do not fully understand the homo- or heteroscedasticity, we believe that this method will not pose a problem within the dynamic range of analysis using MS.

In figure 4C, abrupt changes were due to the instrument maintenance. The reason why this phenomenon did not occur with FA18:1 was unknown. We have added the following sentence in the main text.

P9 Line 275

In addition, abrupt changes were observed at around 4,000, 7,000, and 10,000 chromatographic runs, which correspond to annual instrument maintenance.

4) Conclusion.
I would recommend including a paragraph mentioning the limitations of this study, such as the fact that, from what I understand, only one analytical apparatus was used. In other words, it will be interesting in the future to involve more than one laboratory and/or more than one instrumentation. Indeed, in a clinical application perspective, it will be necessary to correct the values found between different laboratories.

According to the reviewer’s comment we have added the following sentence in the main text.

P13 Line 344

In this study, all samples were analyzed using a single instrument. The usefulness of this method should be verified in the future by examining multiple apparatus, apparatus from verious vendors, and multiple laboratories.